# How to Select Balance Measures Sensitive to Parkinson’s Disease from Body-Worn Inertial Sensors—Separating the Trees from the Forest

**DOI:** 10.3390/s19153320

**Published:** 2019-07-28

**Authors:** Naoya Hasegawa, Vrutangkumar V. Shah, Patricia Carlson-Kuhta, John G. Nutt, Fay B. Horak, Martina Mancini

**Affiliations:** Department of Neurology, Oregon Health & Science University, Portland, OR 97239-3098, USA

**Keywords:** feature selection, balance dysfunction, objective measures, Parkinson’s disease, inertial measurement unit, wearable technology

## Abstract

This study aimed to determine the most sensitive objective measures of balance dysfunction that differ between people with Parkinson’s Disease (PD) and healthy controls. One-hundred and forty-four people with PD and 79 age-matched healthy controls wore eight inertial sensors while performing tasks to measure five domains of balance: standing posture (Sway), anticipatory postural adjustments (APAs), automatic postural responses (APRs), dynamic posture (Gait) and limits of stability (LOS). To reduce the initial 93 measures, we selected uncorrelated measures that were most sensitive to PD. After applying a threshold on the Standardized Mean Difference between PD and healthy controls, 44 measures remained; and after reducing highly correlated measures, 24 measures remained. The four most sensitive measures were from APAs and Gait domains. The random forest with 10-fold cross-validation on the remaining measures (n = 24) showed an accuracy to separate PD from healthy controls of 82.4%—identical to result for all measures. Measures from the most sensitive domains, APAs and Gait, were significantly correlated with the severity of disease and with patient-related outcomes. This method greatly reduced the objective measures of balance to the most sensitive for PD, while still capturing four of the five domains of balance.

## 1. Introduction

Parkinson’s disease (PD) is the second most common neurodegenerative disease, with an overall prevalence of 400–1900 per 100,000 people (60 years and older) [1,2], resulting in significant mobility decline, loss of independence and increased falls [3]. Postural instability, including balance and gait dysfunction, is a major contributor to falls in PD [4]. Thus, early detection of balance dysfunction in the clinical setting would be a very useful addition to clinical assessments to prevent falls and to evaluate changes in mobility in people with PD over time.

In a recent review [5], balance dysfunction in people with PD was characterized by five main control systems: (1) postural sway during quiet stance (Sway), (2) automatic postural responses (APRs) to external perturbation, (3) anticipatory postural adjustments prior to gait initiation (APAs), (4) dynamic balance during walking (Gait) and (5) limits of stability (LOS). Several studies have shown that people with PD have increased sway area, velocity and jerkiness when standing with eyes open, but not with eyes closed, compared to healthy controls [6,7,8,9,10]. People with PD tend under-respond to external postural perturbations such as the pull test, a push and release test or while standing on moving surfaces, resulting in several slow, small steps to recover equilibrium, instead of one quick, long step [11,12,13]. APAs, in preparation for step initiation, consist of a shift in body’s Center of Pressure (CoP) toward the stance leg and backward to allow the Center of Body Mass (CoM) to move forward and unweight the stepping leg [5]. We have shown the magnitude of APAs for step initiation are reduced in PD, resulting in small APAs with slow, short first step execution [14,15]. Gait dysfunction in people with PD is typically represented as slower gait speed, smaller stride length, longer double support time and increased variability of gait cycle duration [16,17]. In addition, a reduced range of motion (ROM) of trunk and arm swing has been shown as marker of early PD [18]. Turning while walking in people with PD is slower with an increased number of steps and longer duration compared to healthy elderly [19]. LOS, measured by the maximum displacement of the body CoM during leaning that can be controlled without a fall or step [13], are reduced in people with PD [20,21,22]. Although balance dysfunction in people with PD can be organized into five different balance domains, we do not know which domains are the most affected. This knowledge could focus therapeutic interventions and serve as outcomes for clinical trials.

In the past decade, inertial measurement units (IMUs) have become widely used due to their portability; the devices are small, lightweight and lower-cost compared to the traditional biomechanics systems used in a gait analysis laboratory [23]. Previous studies using IMUs have shown that people with PD had reduced stride length, foot clearance and increased stance phase time while walking compared to healthy control subjects, similar to the measures collected by a force plate or a motion capture system [24,25]. Recently, we developed algorithms to quantify postural dysfunction during clinical tests of all balance domains using IMUs attached to the feet, trunk and wrists [26,27,28,29]. Work from our laboratory and others demonstrated and validated novel methods to objectively characterize sway, APAs, gait, turning and APRs in healthy older adults and people with PD by using wearable IMUs [8,26,28,30,31,32,33,34,35,36,37,38]. Most previous studies quantified various domains of postural control using a force plate, EMG or a motion capture system, but these systems require long preparation and pre-processing time, as well as skilled personnel and therefore are difficult to apply in the clinical setting. Use of body-worn IMUs to quantify postural control is fast and easy but results in a plethora of outcome measures. However, it is unclear which objective measures from all five balance domains are the most sensitive to detect balance dysfunction in people with PD compared to healthy control subjects. Hence, the purpose of this study is to identify a small set of sensitive objective measures of balance dysfunction that differ between people with PD and healthy controls for use in future rehabilitation interventions.

## 2. Materials and Methods

### 2.1. Participants

A total of 144 subjects with idiopathic PD and 79 age-matched control subjects (free of any neurological or musculoskeletal disorders) participated in this study. Inclusion criteria for people with PD were: (a) age between 50–90 years old, (b) no major musculoskeletal or peripheral disorders (other than PD) that could significantly affect their balance and gait, (c) ability to stand and walk unassisted, (d) meet criteria for idiopathic PD according to the Brain Bank Criteria for PD [39] and (e) no recent changes in medication (six weeks of stable medications). Exclusion criteria for both groups were: any other neurological disorders or musculoskeletal impairments that interfere with gait or balance, and inability to follow instructions.

### 2.2. Procedure

Subjects with PD were tested in their practical Off state after at least 12 h of antiparkinsonian medication wash-out. After obtaining informed consent, the following clinical tests were administered: (1) Movement Disorder Society-sponsored revision of the Unified Parkinson’s Disease Rating Scale (MDS-UPDRS) [40], (2) the Activities-specific Balance Confidence scale (ABC-scale) [41], (3) the Montreal Cognitive Assessment (MoCA) [42] and (4) the Parkinson’s Disease Questionnaire-39 (PDQ-39) [43].

To quantify balance control, the subjects wore eight IMUs (Opals, APDM) that included triaxial accelerometers, triaxial gyroscopes and magnetometers recording at 128 Hz. The sensors were attached to both feet, shins, wrists, sternum and the lumbar region. Subjects performed a total of 10 different motor tasks to characterize the following balance domains: Sway, APRs, APAs, LOS and Gait [20]. In addition, gait and step initiation tasks were performed with and without a concurrent cognitive task (Single task: ST, Dual Task: DT, respectively). Most of the motor tasks were collected while undergoing the clinical assessment of the Mini Balance Evaluation System Test (Mini-BEST) [44]. This study was part of a larger interventional study [45] where participants repeated the same assessment after an education and exercise intervention of six weeks each.

Sway: subjects were instructed to stand quietly for 30 s in four different conditions; firm surface with eye open or closed (EOFirm or ECFirm), and foam surface with eye open or closed (EOFoam or ECFoam). In EO condition they were asked to look at an art poster 6 m ahead. In all sway tasks, hands were kept on hips and feet together until almost touching.

APRs: the Push and Release maneuver was used to measure automatic postural responses to an external perturbation. Subjects were asked to stand still with arms at their sides with their feet should width apart. Subjects then leaned backward against the tester’s hands beyond their backward base of support. They were instructed to do whatever was necessary to regain balance, including taking one or more steps, to avoid falling when the tester quickly removed support. We used a template to achieve consistent foot placement with 10 cm between heels and a 30° outward rotation of the feet in the Push and Release, Instrumented Stand and Walk test (ISAW, below) and LOS (below) [46].

APAs: subjects performed the Instrumented Stand and Walk test (ISAW in Mobility Lab by APDM) with and without a cognitive task (DT and ST). The ISAW consisted of subjects standing without moving for 30 s, followed by a verbal instruction to initiate walking straight ahead for 7 m, turn 180° after crossing a line on the floor, and return to the initial starting location [28]. During quiet standing, they were asked to keep their arms at their sides and look straight ahead. In the DT condition, they were instructed to perform serial subtraction by threes from a three-digit number during both the quiet stance and walking part of the ISAW task. The standing phase and the step initiation phase were considered from this task (Sway and APAs).

LOS: to measure forward-backward limits of stability, subjects were asked to stand still with arms at their sides for 5 s after which they were asked to lean forward as far as they could at a comfortable speed, hold the maximum position for at least 5 s and then lean backward as far as possible and hold 5 s, and then return to the upright stance position. Subjects were instructed not to keep their feet flat on the floor and torso straight.

Gait: to quantify gait characteristics, subjects were instructed to walk at a comfortable pace back and forth continuously between two lines 7.62 m apart for 2 min in the ST condition. This walking task was repeated in the DT condition (reciting every other letter of the alphabet). The ST condition was always completed before the DT condition.

### 2.3. Outcome Measures

Analysis focused on seven measures of Sway, seven measures of APRs, six measures of APAs, 12 measures of Gait and three measures of LOS illustrated in Figure 1 (see also Table A1) based on prior knowledge about test-retest reliability and validity [26,27,28,29]. These objective measures were partly extracted automatically with APDM proprietary software (Mobility Lab version 2) and partly calculated from custom MATLAB algorithms. Specifically, the objective measures of Sway and Gait were computed online with APDM’s Mobility Lab^TM^ (The APDM Inc., Portland, OR, USA, http://apdm.com). The sway measures were extracted from the Opal accelerometer placed on the lumbar region and the gait spatio-temporal parameters were extracted by the orientation and position, estimated using Unscented Kalman Filter to fuse data from Opal accelerometer, gyroscope and magnetometer of the sensors placed on the feet. Measures characterizing APAs, APRs and LOS were calculated offline by using previously validated custom-made algorithm in MATLAB R2018b (The Mathworks Inc., Natick, MA, USA). For example, APRs were characterized with Latencies and Length of first step, Time to stability and Number of steps [26] from the IMUs placed on the feet and on the lumbar region. APAs were characterized by amplitude of the preparatory phase and first step characteristics from the sensors on the shins and lumbar region [38]. Functional LOS was calculated by using estimated COP displacement in AP direction for the LOS task (Forward, Backward and Total range). We estimated COP displacement from the acceleration of the lumbar sensor by using an inverted-pendulum model [47].

A total of 93 objective balance measures were collected in this study. In addition, we calculated the dual task cost (DC), for spatial gait measures of Gait speed and Stride length, as well as temporal gait measures of Stance time, Double support time and standard deviation (SD) of Gait cycle duration. Dual task cost was calculated as:DC [%] = 100 * (DT metric − ST metric)/ST metric(1)

The clinical Mini-BEST was also assessed as clinical measure of dynamic balance with four domains (Sway, APRs, APAs and Gait). The total of MDS-UPDRS, and the sub-total of Part II and Part III were used as measures of disease severity. The MoCA score was used as a measure of overall cognition. The ABC scale was used to assess balance confidence and balance self-perception. The PDQ-39 provided a clinical measure of health-related quality of life (QOL) in subjects with PD.

### 2.4. Statistical Analysis

The distribution for each demographic and clinical measures of the two groups was examined by the Shapiro-Wilk test. For data that were non-normally distributed, the Mann-Whitney U test was used to determine a difference between groups. Otherwise, independent samples t-test was used to examine possibly group differences, and Chi-squared test was used to compare the rate of gender between groups.

The ECFoam task (sway, eyes closed on foam) had more than 20% missing values due to excessive difficulty for both cohorts; thus, it was removed from the analysis. We then categorized the missing data into three categories: (A) subjects fell (Fall), (B) the task was skipped because of subject’s fatigue or limited time (Skip), (C) technical difficulties with the raw sensors data (not usable data). In addition, for APAs measures, we added more two categories; (D) no detectable APAs (no APAs), (E) algorithm cannot detect APAs because baseline data was too noisy (Noisy). After finishing categorization, the missing values were replaced by the single value imputation method or the multiple imputation methods [48]. For the category A and D, the single value imputation method was used by replacing the missing value with the worst-case value plus 2SD or the worst case subtracted 2SD. For the category B, C and E, we adapted the multiple imputation method to replace the missing value by using RStudio version 1.1.463. The procedure involved creating 25 separate data sets in which missing values were replaced with predicted values. The following analyses were carried out separately on all 25 data sets and presented results were the mean from these 25 analyses.

To determine the most sensitive objective measures of balance dysfunction, we used both the Standardized Mean Difference (SMD) and Random Forest methods. The SMD between subjects with PD and healthy controls was calculated with the followed formula [49]:(2)$$SMD\;=\;\frac{{\bar{X}}_{1}\;–\;{\bar{X}}_{2}}{S}$$
where ${\bar{X}}_{1}$ and ${\bar{X}}_{2}$ are the sample means of the two independent groups (subjects with PD and healthy controls);

(3)$$S\;=\left(\frac{SD_{1}^{2}}{n_{1}}\;+\;\frac{SD_{2}^{2}}{n_{2}}\right)\times \left(\frac{n_{1}\;\times \;n_{2}}{n_{1}+n_{2}}\right)$$

*SD*_1_, *SD*_2,_ and n_1_, n_2_ are the sample of standard deviation and the number of samples of these groups, respectively. An SMD value of 0.20 represents a small, 0.50 a moderate and 0.80 a large difference between subjects with PD and healthy controls [50]. Measures that had SMD value greater than 0.50 were retained as sensitive objective measures. Then, correlations between each objective measure were assessed using Spearman’s rho correlations, and measures with a correlation higher than *r* = 0.70 [51] were removed to avoid multi-collinearity in accordance with followed rules: (1) measures with smaller SMD value (to remove the measures less sensitive to PD) and (2) measures calculated for dual tasks (to simplify task performance).

The random forest algorithm used an ensemble learning method by constructing a collection of trained decision trees, was used to find a subset of highly sensitive objective measures and to compare the result with that using SMD [52]. First, a trained tree was created from a random sampling on the data set itself. Once a tree was constructed, a set of data, which did not include any particular record from the original dataset was used as a test set, and the error rate of the classification of all the test sets was calculated (also called out-of-bag error). The concept of variable importance was assessed by Feature of importance. The Feature of importance is a measure of prediction power of variables in regression or classification, and calculated by the Gini index formula [53]:(4)$$Feature\;of\;importance\;\left(n\right)=1-\sum_{j=1}^{2}{\left(p_{j}\right)}^{2}$$
where *p_j_* is the relative frequency of metric *j* in the data set *n*. To validate the method, we used a randomized 10-fold cross-validation [54,55,56,57]. For 10-fold cross-validation, the data are split such that 90% of randomly selected recordings are used for training, whereas the remaining 10% are used for validation.

In addition, we tested the association between the sensitive objective balance measures with the clinical dynamic balance (Mini-BEST), the balance confidence (ABC scale), QOL (PDQ-39 (Total and Mobility sub-score)), and severity of disease (MDS-UPDRS Total, Part II and Part III) using Spearman’s Rho correlations. For the Mini-BEST and the ABC scale, the association was examined in the entire subject group including both PD and control groups, whereas the association with other clinical measures was tested only in the group with PD. A false discovery rate adjustment was used to account for multiple comparisons. The statistical analysis for the demographic data, clinical measure and correlation between clinical measures and objective measures, and the calculation of ICC were processed using SPSS Statistics version 25.0 (IBM, Armonk, NY, USA). The statistical significance was set to *p* < 0.01.

Lastly, the minimal detectable change (MDC) of highly sensitive objective measures in people with PD was calculated as:(5)$$\mathrm{MDC}\;=\;\mathrm{SEM}\;1.96\sqrt{2}$$
where SEM is the standard error of measurement [58]. SEM was calculated as follows:(6)$$\mathrm{SEM}\;=\;\mathrm{SD}\;\sqrt{\left(1-\mathrm{ICC}\right)}$$

SD is the sample of standard deviation and ICC is the intraclass correlation coefficient of measures. To calculate ICC, the objective measures were collected from people with PD in a re-test section which examined after a six-week education intervention.

## 3. Results

Age, height, weight, gender and MoCA scores were not different between the PD and control groups. However, subjects with PD had significantly lower Mini-BEST and ABC scale scores compared to healthy control subjects (Table 1).

Figure 1 shows the methodology and sequential reduction of measures using feature selection. Only the ECFoam postural sway task was removed from analysis due to missing values associated with inability of a large percent of subjects to stand for 30 s (PD 38.9%; HC 11.4%). The EOFoam postural sway task also had missing values due to task difficulty (PD 13.9 %; HC 2.5%), APR (PD 13.9%; HC 1.3%) and APA DT (PD 12.5%; HC 5.1%), but missing values were imputed. After imputation, and a threshold on SMD (>0.5) between PD and control groups, 44 out of 93 measures remained (see Table A2).

Then, the cross-correlation matrix showed 20 of 44 objective measures had significant correlations with measures within the same domain of postural control (threshold = 0.7; dark red and blue in Figure 2). After reducing highly correlated measures, 24 measures were selected as highly sensitive, nonredundant, objective measures of mobility dysfunction in people with PD compared with control subjects (Figure 3A).

Four of five domains of balance (Sway, APRs, APAs and Gait) were sensitive to PD with moderate-to-strong effect size. Measures within the Gait domain were the most sensitive to PD, followed by APAs, then Sway and then APRs. Figure 3 shows the results of SMD and random forest analysis for the top 24 measures that discriminated task performance between people with PD and healthy control subjects. The objective measures with the highest feature of importance using the random forest approach were very similar to those with the highest SMDs. The top four most sensitive objective measures in SMD were ranked the highest in the random forest: Turn velocity (ICC = 0.95, MDC = 23.26), Foot strike angle (ICC = 0.97, MDC = 2.93) and Arm ROM (ICC = 0.96, MDC = 6.44) in Gait ST, and First step ROM in APA ST (ICC = 0.82, MDC = 10.94). In addition, running the random forest with 10-fold cross-validation on the selected 24 measures showed an accuracy of 82.4 ± 12.0 % (mean ± SD) and a precision of 72.5 ± 39.1 % (mean ± SD) compared to an accuracy of 82.4 ± 13.7 % and a precision of 70.8 ± 39.7 % on all 86 measures except measures in ECFoam. ICC and MDC of all 24 sensitive measures are shown in Table A3.

The Spearman’s Rho correlations showed a significant association of both the ABC scale and the Mini-BEST with the objective measures of all four sensitive balance domains (Figure 4A,B). The Spearman’s Rho correlations showed a significant correlation between the clinical measures of QOL and disease severity, and the sensitive objective measures in people with PD (Figure 4C,D). The total of PDQ-39 significantly correlated only with Gait measures; however, the mobility sub-score significantly correlated with the measures in both the Sway and Gait domains. Finally, the total of MDS-UPDRS score correlation pattern with the objective measures was very similar to correlations with the ABC scale except the MDS-UPDRS score showed no correlation with APRs measures, unlike the ABC. Moreover, the MDS-UPDRS Part II score showed significant correlation with Sway and Gait measures, and the Part III score significantly correlated with APAs and Gait measures. Interestingly, there were no significant correlations between Sway or APRs measures and the Motor sub-score (Part III) of MDS-UPDRS.

The four most sensitive objective measures including APAs and Gait domain showed higher correlations with all clinical measures except the total of PDQ-39. The highest correlations among objective measures and clinical measures were for the following (Figure 5); Foot strike angle in Gait ST and the Mini-BEST score, the ABC scale, the mobility score of PDQ-39 and the MDS-UPDRS Part II (*r* = 0.64, *r* = 0.61, *r* = 0.40 and *r* = −0.37, respectively), and Turn velocity in Gait ST with the total of MDS-UPDRS and Part III (*r* = −0.43 and *r* = −0.51, respectively).

## 4. Discussion

Our findings showed that out of 93 objective measures of five different balance domains, 24 objective measures within four main balance domains (Sway, APRs, APAs, and Gait) were identified as the most sensitive, uncorrelated measures of balance dysfunction in PD. Furthermore, these 24 sensitive measures also showed clinical validity as they were significantly related to balance confidence, quality of life and disease severity in people with PD. To our knowledge, this is the first study to determine the most sensitive objective balance measures across five different balance domains to discriminate between people with PD and healthy control subjects. It is also the first study to demonstrate how IMUs can be used to quickly quantify a large set of clinical tests of postural control.

All 24 sensitive measures are consistent with the characteristics of APA, APRs, gait and postural sway dysfunction in people with PD reported in previous studies [8,9,10,11,12,13,14,15,16,17,18,19,24,25,30,31,32,33,34,35,36,37,38], with SMD (>0.5). Only the measures in the LOS domain were not included as sensitive measures as showed by a small difference between people with PD and healthy controls (SMD < 0.5) although a previous study showed forward LOS to be smaller than in people with advanced PD than controls [21]. Turning (Turn velocity) and the pace component of gait (Foot strike angle, Arm ROM and Gait speed) showed the strongest difference between people with PD and healthy controls [59]. Previous rehabilitation and levodopa studies have reported improvements of Gait speed and Stride length and Arm swing, but not of Stance time [60,61]. These results may indicate that the pace components of gait, but not temporal components, are more impaired and associated with bradykinetic symptoms of PD and therefore may be more amenable to therapy than the temporal components.

The second most sensitive balance domain to distinguish people with PD and healthy controls was APAs. As expected, Peak acceleration in ML direction and First step ROM during step initiation were smaller in people with PD than controls, as previously reported by our group and others [15,31,37,62]. Interestingly, our results found slightly higher sensitivity of balance dysfunction of PD by measuring First step ROM (approximation of first step length) rather than Peak acceleration in ML direction, whereas our group previously showed sensitivity to PD only for Peak ML acceleration of APA and not First step parameters in a group of early, untreated PD [31]. This may be explained by the fact that in the same cohort straight ahead gait did not differ compared to healthy elderly individuals [18], while postural preparation was impaired early in the disease prior to start levodopa medication. It is possible that when gait starts deteriorating with disease progression, the first step characteristics deteriorate initially, potentially making gait parameters a good marker to track disease progression while postural preparation would be more sensitive to early disease [32]. Supporting this concept, our correlation results showed the motor part of MDS-UPDRS moderately correlated with First step ROM, but not with Peak ML APA. These results may indicate that rigidity and bradykinesia affect the size of first step rather than that of APAs. Furthermore, in the present study, APAs were measured within the ISAW task, which may increase APAs in people with PD. In fact, in the ISAW task, participants initiated a step followed by tester’s instruction to “Walk”, which may be an auditory cue that can increase the size of APAs, but not improved steps in people with PD [63].

Following the Gait and APAs domains, the measures of the Sway domain showed the next highest sensitivity of PD. The most sensitive measures of balance dysfunction of PD in the Sway domain were the root mean square (RMS) of acceleration in both ML and AP directions during standing on a foam surface with eyes open. In addition, the number of measures highly sensitive to PD (SMD >0.5) included two measures in EOFirm condition, seven measures in the EOFoam condition, and one metric in the ECFirm condition. These findings may indicate that the task, standing on a foam surface with eyes open, is the most effective condition to detect a postural sway dysfunction in PD. In fact, a previous study showed people with PD increased the risk of falls under the unstable surface condition, but not with eyes closed [64]. Previous studies have shown the impairment of proprioceptive sensory integration in both peripheral and central sensory systems [65,66,67]. For example, a recent study showed more severe PD patients tend to weight to proprioceptive information rather than vestibular information even on a perturbed surface compared with healthy controls [68]. This finding may indicate that people with PD tend to use proprioceptive information for stabilizing balance, even though they cannot detect precise information from a foam surface. Interestingly, our findings did not show a strong sensitivity in Jerk measures, as previously reported by our group [8,28]. This different finding between previous studies and our findings may be because prior studies tested untreated people with PD, or people on levodopa medication, but in our study, our participates were tested Off medication. In fact, sway Jerk is higher in early, untreated people with PD compared to healthy controls, but, as the disease progresses, the Jerk becomes lower than controls [32]. Moreover, levodopa medication tends to decrease Jerk measurements. These results may indicate that a lower than normal Jerk is associated with reduced postural corrections in PD in a moderate to severe stage, as opposed to “trying too much” to correct posture at an early stage of the disease. Thus, variation in disease severity among subjects will increase the variability of jerkiness among subjects with PD, which may reduce its effect size.

The APRs showed the least sensitivity among the most sensitive postural control domains of PD patients. As consistent with the previous study using the push and release maneuver [69], the most sensitive measures in APRs were the Length of compensatory stepping, but the effect size was moderate (SMD <0.8). Interestingly, our objective measures of APRs appear to be more sensitive than clinical measures of APRs. Stepping responses were significantly small, even though most of the PD participants (more than 80%) were less than stage 3, as assessed by the Hoehn and Yahr stage [70], which indicates the subject does not show apparent clinical instability in responding to the pull test.

LOS did not indicate high sensitivity to discriminate PD patients and the healthy elderly. Previous studies showed that people with PD reduced their LOS compared to healthy controls even in their ON state [21,22], but this was not supported in our findings. These different findings may be explained by subjects in other studies having more advanced stage or longer disease duration compared to our cohort of PD. Consistent with our results, a recent study showed PD subjects of the Hoehn and Yahr stage around 2 and with 4.5 years from diagnosis of PD did not reduce their LOS compared with healthy controls [71]. Another explanation for the difference may be that a different formula to calculate LOS; for our findings, we estimated COP displacement from the acceleration of the lumber sensor by assuming an inverted-pendulum, but for the previous study, they calculated COP displacement by using force plates. In fact, people with PD cannot lean their bodies maximally strictly as inverted pendulums and may show more flexion at hips and knees compared to healthy controls which would reduce the measured LOS [21].

This study also provides ICC and MDC values for the sensitive measures. The ICC represents the test-retest reliability estimates, and the MDC represents the minimum amount of change that must take place for the difference to be considered “real” rather than measurement error [58,72]; thus, these values are very useful for clinical assessment. The top four sensitive measures showed good reliability (ICC >0.82). Furthermore, the results seem to suggest the reliability of measure may decrease as the SMD are lessened (see Table A3). This may indicate that using the higher sensitive measures of PD dysfunction compared to healthy controls are important to detect the changes more precisely in postural control systems.

The top four balance measures within the Gait and APAs domains had both a larger SMD and higher feature of importance using a random forest algorithm. All four most sensitive objective measures of PD were highly associated with all clinical measures. Consistent with the present study, our previous study showed a correlation between balance confidence, motor function of QOL and the clinical disease severity in motor function and Gait stride velocity, Turn measurements and First step ROM in people with PD using the ISAW task [73]. In addition, both studies reported the highest correlation between motor UPDRS part III with the Turning measures, which was the most sensitive measure to detect the balance dysfunction of PD calculated by SMD. Furthermore, our results showed the measure of Foot strike angle, which was the second most sensitive measure indicated by SMD was also highly correlated with the clinical dynamic balance, the perceived balance confidence, the motor function or mobility of QOL and the ADL impairment, as assessed by MDS-UPDRS Part II. Thus, all four most sensitive measures, particularly Foot strike angle and Turn velocity, may not only represents the characteristics of balance and gait dysfunction in people with PD, but also be useful to assess the clinical dysfunction by clinicians.

There are several limitations to this study. One limitation is the study design treating missing values. The objective measures in ECFoam had too many missing values to do our analysis but may be more sensitive rather than other sway conditions to detect balance dysfunction in people with PD. In fact, only 60% of PD subjects kept standing more than 30 s on a foam surface with eyes closed, while about 90% of HC subjects could keep standing in similar conditions. Another limitation is the disease severity of PD in our subjects. Most of our cohorts of PD were mild severity assessed by the Hoehn and Yahr stage, which may underestimate the balance dysfunction of PD, especially in the LOS domain. Lastly, it is possible the fixed order of the tests and conditions may have introduced an order effect. However, most of objective measures collected in DT or complex condition (EC or Foam conditions) had higher SMD even though these complex conditions were performed after a simple condition.

## 5. Conclusions

This study illustrates how a systematic statistical procedure can reduce the number of potential objective measures of balance dysfunction needed to differentiate people with PD from healthy controls. The proposed method reduced measures from 93 to 24 independent measures that still obtained similar classification accuracy. The four most sensitive objective measures characterized gait and gait initiation and correlated very well with clinical measures of dynamic balance, perceived balance confidence, QOL, disease severity and ADL disabilities. These findings suggest that clinical trials focused on improving balance dysfunction in people with PD should use these four measures: Turning velocity, Foot strike angle, Arm swing ROM and First step length.

## Figures and Tables

**Figure 1 sensors-19-03320-f001:**
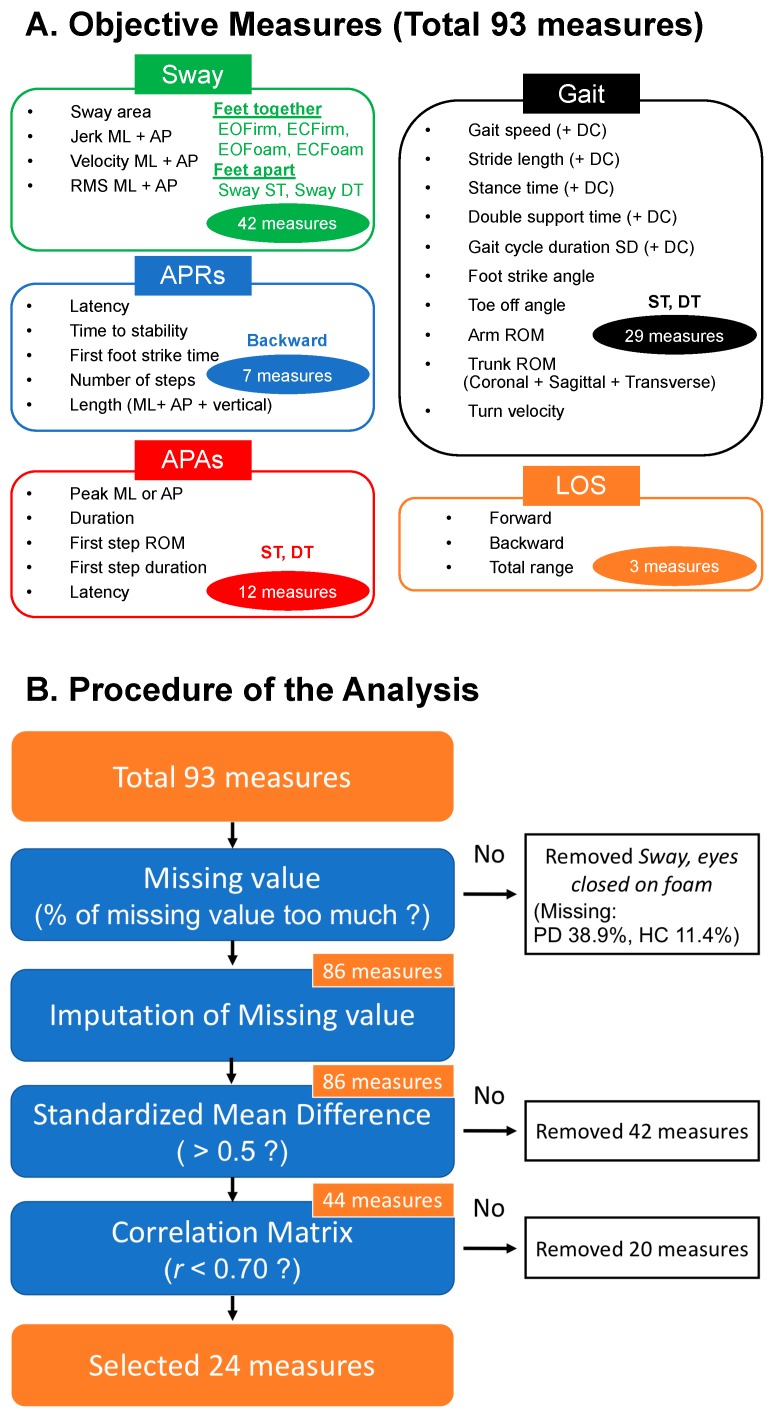
Methodology of the analysis. (**A**) Total objective measures measured from various mobility tasks. For the Sway domain, six tasks were performed. Both dynamic posture (Gait) and anticipatory postural adjustments (APAs) included two tasks and both automatic postural responses (APRs) and limits of stability (LOS) domains had one task. We measured a total of 93 objective measures. (**B**) Procedure of the analysis. The orange colored squares indicate the number of the remaining measures. 24 objective measures selected from 93 measures by the proposed analyses. ST, single task; DT, dual task; DC, dual task cost; EOFirm, eyes open on firm; EOFoam, eyes open on foam; ECFirm, eyes closed on firm; ECFoam, eyes closed on foam; RMS, root mean square of acceleration.

**Figure 2 sensors-19-03320-f002:**
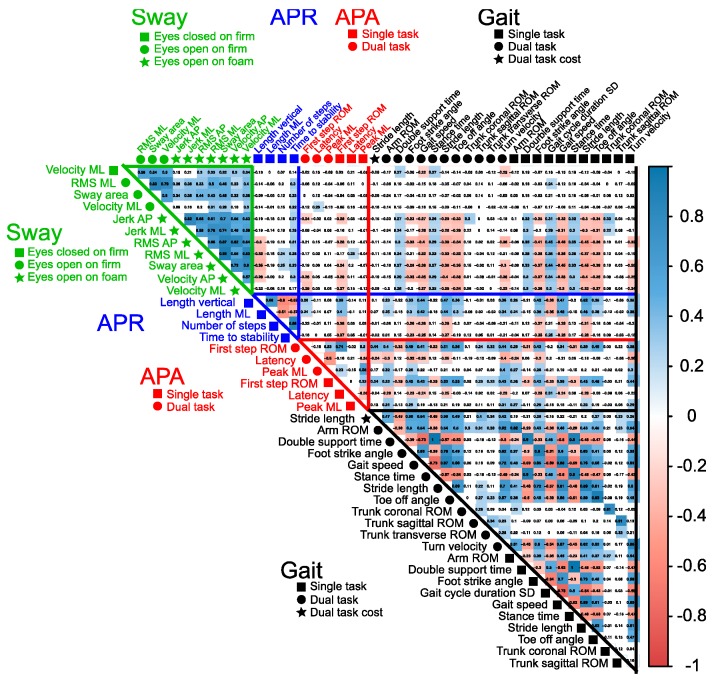
Correlation matrix of sensitive measures. The 44 sensitive measures passed the criteria of Standardized Mean Difference (SMD) between PD subjects and healthy controls (SMD > 0.5). The shape with the name of the measures shows each task. A blue colored cell indicates a positive correlation between the measures, whereas a red colored cell indicates a negative correlation between the measures. The darker a cell is colored, the higher the correlation becomes. RMS, root means square.

**Figure 3 sensors-19-03320-f003:**
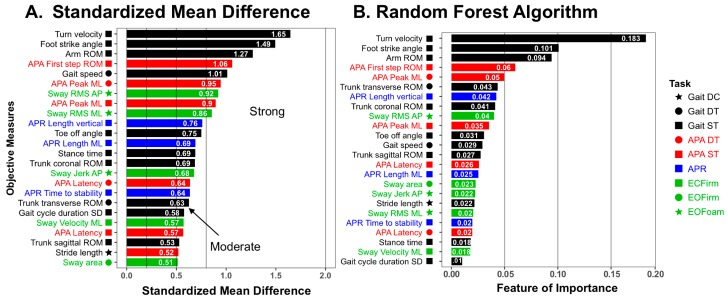
Bar plot of the Standardized Mean Difference (**A**) and Feature of Importance analyzed by the Random Forest Algorithm (**B**) in the sensitive 24 measures. Absolute values are given in the SMD plot. The top of the four measures shows both higher SMD and higher feature of importance. ST, single task; DT, dual task; DC, dual task cost; EOFirm, eyes open on firm; EOFoam, eyes open on foam; ECFirm, eyes closed on firm; RMS, root means square.

**Figure 4 sensors-19-03320-f004:**
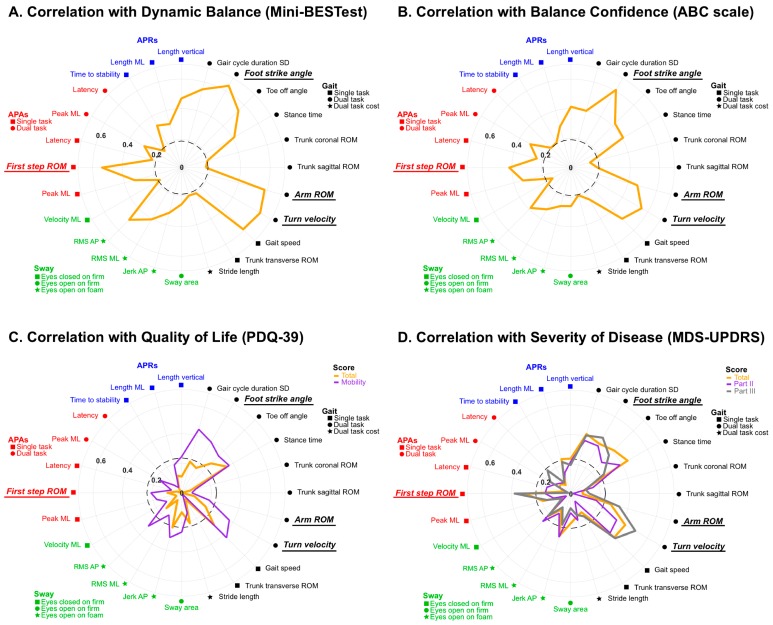
Polar plot comparing Spearman’s rho correlation of (**A**) the Mini Balance Evaluation Test (Mini-BEST), (**B**) the Activities-specific Balance Confidence scale (ABC-scale), (**C**) the Parkinson’s Disease Questionnaire (PDQ-39) and (**D**) the Movement Disorder Society-sponsored revision of the Unified Parkinson’s Disease Rating Scale (MDS-UPDRS) with the sensitive 24 objective measures in each domain of clinical measures. Absolute values are given. The black dashed circle indicates the threshold of a significant correlation.

**Figure 5 sensors-19-03320-f005:**
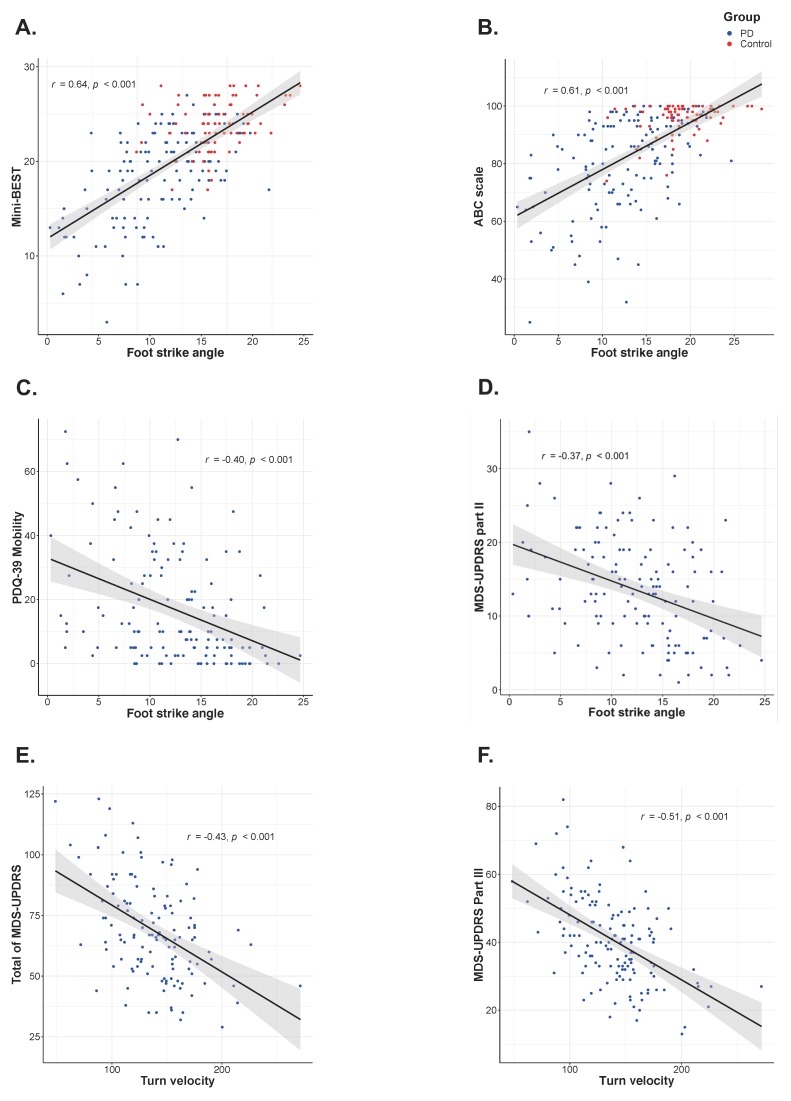
Scatter plot of the metric with the strongest correlation within the four most sensitive measures with (**A**) the Mini Balance Evaluation Test (Mini-BEST), (**B**) the Activities-specific Balance Confidence scale (ABC-scale), (**C**) the Mobility sub-score of Parkinson’s Disease Questionnaire (PDQ-39 Mobility), and the Movement Disorder Society-sponsored revision of the Unified Parkinson’s Disease Rating Scale (MDS-UPDRS) Part II (**D**), Total score (**E**) and Part III (**F**) assessed by Spearman’s rho correlation.

**Table 1 sensors-19-03320-t001:** Demographic data.

	Controls (N = 79)		PD (N = 144)		*p* Value
	Mean	SD	Mean	SD	
Male/Female	48/31		93/51		0.571 ^a^
Age	68.2	8.1	68.4	8.0	0.822
Height (cm)	171.5	10.0	173.4	9.9	0.184 ^b^
Weight (kg)	73.7	13.1	79.7	16.9	0.018 ^b^
Disease Duration (years)	-	-	6.2	5.0	-
MDS-UPDRS					
Total	-	-	68.7	20.4	-
Part II	-	-	13.6	7.0	-
Part III	-	-	40.6	12.6	-
Mini-BEST	24.0	2.6	18.5	4.8	**<0.001 ^b^**
ABC scale	95.9	5.3	80.5	16.3	**<0.001 ^b^**
PDQ-39					
Total	-	-	17.6	11.8	-
Mobility	-	-	17.0	17.4	-
MoCA	26.8	2.3	25.8	3.4	*0.080* ^b^
Hoehn & Yahr stage	-		1/115/15/13		-
(I/II/III/IV)					

Groups compared using independent sample *t*-test, Mann-Whitney U test or Chi-squared test and significance level of 0.01 (^a^: Chi-squared test, ^b^: Mann-Whitney U test). Bold values indicate significant differences between groups (Control and PD). PD, Parkinson’s disease; MDS-UPDRS, Movement Disorder Society-Sponsored Revision of the Unified Parkinson’s Disease Rating Scale; Mini-BEST, Mini Balance Evaluation Systems Test; ABC scale, the Activities-specific Balance Confidence scale; PDQ-39, Parkinson’s Disease Questionnaire-39; MoCA, Montreal Cognitive Assessment; SD, Standard Deviation.

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
