# Peer review of "How to Select Balance Measures Sensitive to Parkinson’s Disease from Body-Worn Inertial Sensors—Separating the Trees from the Forest"

_sensors, 2019, doi:10.3390/s19153320_

Round 1
Reviewer 1 Report
It is a very interesting paper and very well written. This paper deserves to be published. The methodology is rigorous and well explained. The results are very clear. I congratulate the authors for this remarkable work. There is a lack of studies that use Imus in a clinical context. I would just like the authors to add a paragraph on the processing of the Imus signal to allow other teams to learn from their work.
Did the authors calibrate the Imus? if so, with a functional calibration? What signal did the authors process? Accelerometers? Gyroscopes? How did the authors deal with the problem of signal drift? Did the authors use the magnetometer?
Reviewer 2 Report
Authors evaluated the most sensitive objective measures related to balance dysfunction useful to discriminate between healthy subjects and patients with PD. The aim of the paper is clearly stated and the topic is interesting for the research field. However, manuscript should be improved in order to increase the scientific soundness and the quality.
Please consider these remarks to solve:
INTRODUCTION:
Since the data were gathered via IMU, I suggest to enrich the Introduction with a paragraph on the use of IMU for balance assessment.
I suggest to enrich the comments related to the novelty of your paper with respect the ones already published in literature
I suggest to add a short paragraph related to the idiopathic PD considering the type of patients enrolled in the study.
Please add a reference for the sentence on pros of IMU with respect the other sensor systems.
METHODS:
Since you computed parameters related to the ROM, I suggest to add an explanation on how the placement of the sensors was performed. In fact, it is well-know that a misalignment could provide wrong estimation of the joint angles.
Why did you decide to perform the sway test by reducing the base of support? "feet together until almost touching", please justify.
Were the 10 tasks randomized among patients? Please clarify and justify; in fact if no randomization were performed a bias could be affected the data. Please justify also why ST condition was always completed before DT one.
I strongly suggest to discuss all the computed parameters by explain each one in details with formula and meaning.
Please report a reference related to the r range in order to understand if the correlation is strong, moderate, modest and so on (please see Germanotta et al. Spasticity measurements based on tonic stretch reflex threshold in children with cerebral palsy using pedianklebot.
I suggest to compute also the precision of the classification and not only the accuracy (error rate) in order to take into account also the influence of false positive rate that is fundamental in clinics.
Please explain how to understand the feature of importance values. Are there some range of values? If no, how can you assess that only the four mentioned parameters are the best ones?
Please report also in the protocol that the tasks were repeated two times after 6-week
Round 2
Reviewer 2 Report
Authors carefully replied to all my comments. Paper is now ready for the publication.